# Characterization of *Leishmania* Parasites Isolated from Naturally Infected Mammals

**DOI:** 10.3390/ani13132153

**Published:** 2023-06-29

**Authors:** Aroia Burguete-Mikeo, Celia Fernández-Rubio, José Peña-Guerrero, Rima El-Dirany, Leonardo Gainza, Belen Carasa Buj, Paul A. Nguewa

**Affiliations:** 1ISTUN Instituto de Salud Tropical, Department of Microbiology and Parasitology, University of Navarra, c/Irunlarrea 1, E-31008 Pamplona, Spain; aburguetem@unav.es (A.B.-M.); cfdezrubio@unav.es (C.F.-R.); jpena.1@alumni.unav.es (J.P.-G.); reldirany@alumni.unav.es (R.E.-D.); 2IdiSNA (Navarra Institute for Health Research), University of Navarra, c/Irunlarrea 1, E-31008 Pamplona, Spain; 3Laboratory of Molecular Biology and Cancer Immunology, Faculty of Sciences I, Lebanese University, Hadath 1003, Lebanon; 4Clinica Veterinaria Burlada, Plaza Ezcabazabal 2, E-31600 Burlada, Spain; veterinariaburlada@gmail.com; 5Clinica Veterinaria Belen Carasa Buj, c/Ximénez de Rada 53, E-31500 Tudela, Spain; carasavet@gmail.com

**Keywords:** amphotericin B, *APG9*, CYCA, *CYC6*, miltefosine, *Leishmania infantum*

## Abstract

**Simple Summary:**

Leishmaniasis is a group of parasitic diseases that affect humans and animals. Climate change and increased travel and migration have contributed to the spread of leishmaniasis in Europe, which may allow the introduction of new exotic *Leishmania* species or change the profile of known strains. Therefore, it is a priority to continue isolating and characterizing *Leishmania* strains from hosts. In this study, we analyzed and characterized two *Leishmania* isolates (NAV and TDL) obtained from naturally infected mammals (dogs). We identified *Leishmania infantum* parasites, the main agents responsible for the disease in Spain and Europe. We focused on the analysis of growth rate, treatment response, infection capacity, and gene expression, comparing these isolates with the widely studied strain *L. infantum* BCN 150. Considering that these isolates showed different profiles, both NAV and TDL could be useful for in vitro and in vivo assays that might shed some light on the biology of the parasite.

**Abstract:**

Leishmaniasis is spreading in Europe, especially in endemic countries such as Italy and Spain, in part due to ongoing climate change and the increase in travel and migration. Although *Leishmania infantum* is the main agent responsible for this disease in humans and animals, other species and hybrids have been detected. This highlights the need to continue isolating and characterizing *Leishmania* strains from biological samples of infected hosts. In this study, we characterized the recently isolated parasites *L. infantum* NAV and *L. infantum* TDL, obtained from naturally infected mammals (dogs), and we compared them with the widely distributed and studied strain *L. infantum* BCN 150. Both NAV and TDL promastigotes showed a slower growth rate than BCN 150 and were significantly more sensitive to amphotericin B and miltefosine. Furthermore, the expression of the *CYCA* gene (involved in cell cycle and proliferation) was significantly downregulated in NAV and TDL isolates. On the other hand, *CYC6* (implicated in treatment resistance) and *APG9* (related to the recycling of protein under stress conditions and/or while undergoing a differentiation process and treatment resistance) levels were upregulated, compared to those measured in BCN 150. Both isolates displayed a higher infection capacity (>3 amastigotes per macrophage and >70% of infected macrophages) compared to controls (<2 amastigotes/cells and <50% of infected macrophages). Finally, a higher susceptibility to miltefosine treatment was observed in intracellular NAV and TDL amastigotes. In conclusion, TDL and NAV are novel *Leishmania* isolates that might be useful for in vitro and in vivo assays that will allow a better understanding of the parasite biology in Mediterranean areas.

## 1. Introduction

Leishmaniasis is a group of diseases caused by protozoan parasites of the genus *Leishmania*. According to the World Health Organization (WHO), it is considered as one of the 20 neglected tropical diseases [1]. Due to its complex transmission cycle involving humans, domestic animals, wildlife and sand fly vectors, and considering that is a zoonotic and anthroponotic disease [2], it must be approached considering all the aspects that contribute to the disease, from the One Health perspective [3].

In the case of humans, these infections are endemic in 98 countries around the world [4], and mainly affect populations living in poverty [1] in Africa, Asia and South America [2]. Every year about 1 million new cases are reported, and the estimated mortality is more than 13,700 deaths per year [5]. Currently, there is no effective vaccine and the main treatments (such as amphotericin B, miltefosine and pentavalent antimonials) present several disadvantages, such as high cost, severe side effects and drug resistance [6]. There are more than 20 pathogenic *Leishmania* spp. responsible for clinical manifestations: cutaneous leishmaniasis (CL), the most common form of the disease, mucocutaneous leishmaniasis (MCL), post-kala-azar dermal leishmaniasis (PKDL) and visceral leishmaniasis (VL), the most serious form and usually fatal if untreated [1].

Regarding animals, several species of wild, domestic and synanthropic mammals have been recorded as hosts and/or reservoirs of *Leishmania* spp. around the world [7]. In Europe, the most important reservoirs are dogs, which present a broad range of clinical manifestations when ill. Although around 60% of infected dogs remain apparently healthy, they play an active role in the transmission of the disease [8].

Leishmaniasis is spreading in Europe, especially in endemic countries such as Italy and Spain. Foci of the disease are appearing in previously non-endemic European regions. Consequently, several authors consider it an emerging disease in this area [9,10,11,12,13]. It has been demonstrated that ongoing climate change is contributing to the spread of the vector and also accelerating the life cycle of the parasite [13].

Moreover, a questionnaire survey in Europe revealed that leishmaniasis is not notifiable in all countries with autochthonous human and canine cases, and thus the disease continues to be underreported [12]. Currently, prevention, diagnosis, treatment and monitoring of this disease are challenges from the veterinary and public health point of view [8,14]. The main etiologic agent of leishmaniasis in Europe is *Leishmania infantum*, causing VL and CL. It belongs to the *Leishmania donovani* complex, consisting mainly of *L. donovani* and *L. infantum* (*L. chagasi* in the New World) [15]. In addition, other *Leishmania* spp., such as *L. donovani*, *L. major* and *L. tropica* have been reported across Europe [12], as well as the hybrids *L. infantum*/*L. donovani* and *L. infantum/L. major* [13]. These facts, together with the risk of introducing exotic species through international travels or migration [13], reveal the need to continue isolating and characterizing *Leishmania* strains from samples of infected hosts.

In this study, we focused on the isolation and characterization of *Leishmania* isolates obtained from biological samples of naturally infected dogs. We focused our attention on Navarra (Spain), where leishmaniasis is considered hypoendemic [16]. Two *Leishmania* isolates were obtained from biological samples, and after confirming that both belonged to *L. infantum* species, they were named *L. infantum* NAV (NAV) and *L. infantum* TDL (TDL). Different studies were then carried out for their characterization, using as a reference the strain *L. infantum* BCN 150 (MCAN/ES/96/BCN150), which is widely distributed in Spain and commonly used in research labs [17,18,19]. The aim of this project was to provide new characterized *Leishmania* isolates, for their subsequent use in in vivo and in vitro assays. We also aimed to study the drug sensitivity and the infective capability of such isolates to better understand the biology of the parasites that cause the disease in the country.

## 2. Materials and Methods

### 2.1. Parasite Culture and DNA Extraction

The biological samples were initially seeded at 26 °C in Schneider’s Drosophila medium supplemented with 20% heat-inactivated fetal bovine serum (FBS) and an antibiotic cocktail (50 U/mL penicillin, 50 mg/mL streptomycin) in order to facilitate the isolation and growth of the parasites.

Three different *Leishmania* spp., *L. donovani* LV9, *L. infantum* BCN 150 (MCAN/ES/96/BCN150) and *L. guyanensis*, were used as reference and controls. *L. donovani* promastigotes were kindly provided by Philippe M. Loiseau from the Université Paris-Sud, BCN 150 by Prof. Dr. Rafael Balaña Fouce, from the Instituto de Biotecnología de León (INBIOTEC) and *L. guyanensis* were purchased from the American Type Culture Collection (ATCC).

The parasites were collected by centrifugation and the DNA was extracted by following the protocol described by Medina-Acosta et al. [20]. The samples were washed with ethanol and finally purified with RNase A (0.1 mg/mL) for 1 h at 37 °C and proteinase K (0.14 mg/mL) for 1 h at 60 °C. DNA concentrations were measured on a NanoPhotometer NP80 spectrophotometer (Implen, Munich, Germany).

### 2.2. PCR-RFLP

Conventional PCRs were performed in a volume of 50 µL containing 0.5 µM of each primer (*cpb*, Fw 5′ CGTGACGCCGGTGAAGAAT 3′ and Rv 5′ CGTGCACTCGGCCGTCTT 3′) (Hide and Bañuls (2006)) [15], 200 µM of dNTP, 2 U of Taq DNA polymerase, 5 µL of Buffer 10× and 100–200 ng of extracted DNA. Amplification of the *cpb* gene was performed under the following conditions: after an initial denaturation stage of 5 min at 95 °C, 30 cycles were necessary for amplification (denaturation for 1 min at 95 °C, annealing for 1 min at 62 °C and elongation for 1 min at 72 °C), followed by a final elongation of 10 min at 72 °C. Amplicons were analyzed by 1% agarose gel electrophoresis, followed by staining with ethidium bromide and visualization under UV light.

Then, the PCR products were digested for 24 h at 37 °C with the DraIII-HF enzyme (New England Biolabs, Ipswich, MA, USA), according to the manufacturer’s instructions. The results were visualized by agarose gel electrophoresis.

### 2.3. In Vitro Growth Curve of Promastigote Parasites

Growth assays were performed at 26 °C in M199 medium (Sigma-Aldrich, St. Louis, MO, USA) supplemented with 45 mM HEPES (pH 7.2; Sigma-Aldrich), 0.1 mM adenine (Sigma-Aldrich), 0.0005% (*wt/vol*) hemin (Sigma-Aldrich), 0.4% biopterin 0.25 mg/mL (Sigma-Aldrich), 0.001 mg/mL biotin (Sigma-Aldrich), 10% (*vol/vol*) FBS, and an antibiotic cocktail (50 U/mL penicillin, 50 mg/mL streptomycin) (Gibco Laboratories, Paisley, UK). To obtain the growth curve, 10^6^ promastigotes per milliliter were initially seeded in culture flasks, and the parasite concentration was measured daily for 7 days. The results were expressed as parasite number per milliliter ± the standard deviations (SD) from six independent experiments.

### 2.4. Nucleotide Sequences Alignment

Nucleotide sequences data from the *Leishmania cpb* gene were obtained from the National Center for Biotechnology Information (NCBI), Bethesda, MD, USA. Multiple sequences alignments were performed using the BLOSUM62 matrix implemented in the CLUSTALW program.

### 2.5. In Vitro Sensitivity of Promastigotes to Drugs

MTT (3-(4,5-dimethylthiazol-2-yl)-2,5-diphenyltetrazolium bromide) (Sigma, St. Louis, MO, USA) was used to determine the promastigote sensitivity to two drugs currently used as treatment against leishmaniasis. Exponentially growing parasites (5 × 10^6^ promastigotes/mL) were seeded in 96-well plates with increasing concentrations of the drugs (0–500 µM miltefosine (Cymit Química, Barcelona, Spain); 0–2 µM amphotericin B (Acofarma, Barcelona, Spain)) and maintained at 26 °C. Then, 100 µg/well of MTT was added, and the plates were incubated for 4 h under the same conditions. After that, the formazan crystals formed were dissolved with dimethyl sulfoxide (DMSO). The optical density (OD) was measured in a Multiskan FC microplate photometer plate reader at 540 nm, and the half maximal effective concentration (EC_50_) was determined. The EC_50_ represents the concentration required for 50% growth inhibition of treated parasites with respect to untreated parasites (controls). This value was obtained by fitting a sigmoidal *E*max model to dose–response curves. The results were expressed as means ± SD from three independent experiments.

### 2.6. RNA Extraction and Quantitative Reverse Transcription-PCR (RT-qPCR) Gene Expression Analysis

To compare the gene expression profile of different *L. infantum* isolates, 10^6^ parasites/mL were seeded in culture flasks with supplemented M199 medium and incubated at 26 °C during 72 h. Then, the parasites were collected by centrifugation for RNA extraction. Total RNA was purified using the TRIzol reagent (Sigma-Aldrich), as described by the manufacturer. Any possible genomic DNA leftover was removed by treatment with DNase (Ambion Life Technologies, Carlsbad, CA, USA), following the instructions provided by the manufacturer. RNA concentration was measured with a NanoPhotometer NP80 (Implen, Munich, Germany). One microgram of RNA was reverse transcribed using Super Script II Reverse Transcriptase (Invitrogen) to cDNA, which was later used as the PCR template. The PCR was performed in a 96-well plate using a CFX Connect Real-Time System (Bio-Rad Laboratories, Hercules, CA, USA) with SYBR Green PCR master mix (Applied Biosystems, Waltham, MA, USA), according to the manufacturer’s instructions. The *L. infantum glyceraldehyde-3-phosphate dehydrogenase* (*GAPDH*) gene was used as housekeeping to normalize the gene expression of the selected genes (Table 1). The amount of each transcript was expressed by the formula: $2^{CT\left(GAPDH\right)-CT(gene)}$, with the threshold cycle (*CT*) being the point at which the fluorescence rises appreciably above the background fluorescence. The results were obtained from three experiments.

### 2.7. In Vitro Infection of Bone Marrow-Derived Macrophages

BALB/c mouse bone marrow monocytes were differentiated to macrophages using the fibroblast cell line L929 supernatant. Bone marrow-derived murine macrophages were then seeded in 8-well culture chamber slides (Lab-Tek; BD Bioscience, East Rutherford, NJ, USA) at a quantity of 50 × 10^3^ cells per well in supplemented DMEM medium, and allowed to adhere overnight at 37 °C in a 5% CO_2_ incubator. For the infection assay, stationary-phase promastigotes were added at a macrophage/parasite ratio of 1/5 and incubated under the same conditions, allowing macrophage phagocytosis. Twenty-four hours later, wells were washed with preheated PBS to remove extracellular promastigotes, fixed with ice-cold methanol for 5 min, and stained with Giemsa stain. To determine the parasite burden, macrophages and amastigotes were counted under a light microscope until reaching 100 infected macrophages. The percentage of macrophage infection was determined by dividing the number of infected macrophages by the total number of macrophages counted. The number of amastigotes per infected macrophage was calculated by dividing the total number of amastigotes counted by the number of infected macrophages. The experiment was performed in quadruplicate.

### 2.8. Treatment Response of Intracellular Amastigotes

Murine macrophages were seeded and infected as mentioned above. Twenty-four hours after the infection, the cells were treated with 0.05 µM amphotericin B (Acofarma, Barcelona, Spain) or 2 µM miltefosine (Cymit Química, Barcelona, Spain) for 48 h. Then, the wells were washed as described above, and parasite burden was determined. The experiment was performed in triplicate.

### 2.9. Statistical Analysis

The statistical analyses were carried out with PRISM version 5.0 (GraphPad Software Inc., San Diego, CA, USA). Comparisons between isolates in treatment response studies, gene expression studies and infection capacity were performed by Student’s t test. The statistical significance was determined (*** *p*  <  0.001; ** *p*  <  0.01; * *p*  <  0.05). Data were represented as means ± SDs.

## 3. Results

### 3.1. PCR-RFLP Allowed the Specific Species Identification of NAV and TDL Isolates

In this study, the methodology proposed by Hide and Bañuls (2006) [15] and Oshaghi et al., (2009) [21] was used. Hide and Bañuls developed a discriminative PCR for the *L. donovani* complex, focusing on the cathepsin-1 proteases *cpb* gene. These genes have attracted considerable attention, due to their role in the destruction of the host protein and the evasion of its immune response [15,22]. Both species of the complex differ by the length of the fragment. *L. donovani* strains are characterized by a PCR product of 741 bp, while *L. infantum* strains are characterized by a 702 bp product [21]. Specific primers for the *cpb* gene were used in the conventional PCR, and a band of roughly 700 bp size was detected in the two isolates (NAV and TDL), indicating the presence of the *L. donovani* complex *cpb* gene sequence (Figure 1). A band was observed in *L. donovani* and BCN 150 DNA samples used as a positive control, but none in *L. guyanensis* DNA, which was used as a negative control.

The relatively similar length of the *L. donovani* and *L. infantum*
*cpb* gene (a difference of 39 bp) hinders the diagnosis of the species when agarose gel electrophoresis is used for visualization. Oshaghi et al. (2009) proposed the discrimination of these two species by digesting the PCR products with the enzyme DRAIII-HF, whose restriction target is found in the *cpb* gene of *L. donovani* [21]. For that reason, in order to confirm that we were working with *L. infantum* parasites, PCR products were digested with DraIII-HF as described in the Materials and Methods. BCN 150, NAV and TDL PCR products were not digested under tested conditions, indicating a lack of the *L. donovani*-specific 39 bp restriction enzyme cleavage site and confirming that parasites belong to species *L. infantum* within the *L. donovani* complex. The *L. donovani*
*cpb* PCR product was used as a positive control (Figure 2).

### 3.2. PCR Product Sequencing Confirmed That Isolates Were L. infantum

NAV and TDL PCR products were sequenced and aligned with the *cpb* gene of *L. donovani* (AY896783.1) and *L. infantum* (AY896777.1) (Figure 3) using Clustal Omega. Alignment of the sequences confirmed the lack of the 39 bp fragment only present in the *L. donovani*
*cpb* gene.

### 3.3. Isolates Showed a Slower Growth Rate

To analyze the replication rate in vitro of new *L. infantum* isolates, promastigote cultures were initially seeded under the same conditions and parasite concentration was measured periodically. Both NAV and TDL isolates showed a slower growth rate than BCN 150, which was used as a reference (Figure 4). NAV parasites reached the stationary phase after 96 h of incubation, whereas TDL parasites did so after 120 h. On the other hand, BCN 150 reached the stationary phase 72 h after initial culture sowing.

### 3.4. NAV and TDL Promastigote Parasites Were More Susceptible to Miltefosine and Amphotericin B

The sensitivity of the analyzed *L. infantum* isolates to miltefosine and amphotericin B was measured by using the MTT assay, as described in the Methods section. Generated dose–response curves and EC_50_ values are presented in Figure 5 and Table 2, respectively. As the results show, new isolates were more sensitive to the drugs studied, with *L. infantum* TDL being the most susceptible to both treatments. When treated with amphotericin B, TDL exhibited an EC_50_ value of 0.02 µM and near to 9 µM for miltefosine (four times more susceptible than BCN 150 and twice as sensitive as NAV) (Table 2).

### 3.5. The Expression of Several Genes Was Different in the Promastigote Form of the Parasites

Three genes related to treatment resistance, proliferation and autophagy were analyzed by quantitative PCR to determine possible differences between isolates. The *L. infantum* BCN 150 strain was used as a control to normalize our results (Figure 6).

Since both new isolates presented a lower growth rate than the reference strain (Figure 4), we sought to study expression levels of the genes associated with proliferation, such as the *CYCA* gene, which is linked to cell proliferation. *CYCA* levels were markedly downregulated in the NAV and TDL isolates (** *p* < 0.01) (Figure 6), compared to BCN 150.

On the other hand, since both isolates were more susceptible to treatments (Figure 5 and Table 2), we also studied the *CYC6* and *APG9* genes, related to drug resistance and autophagy, respectively. *CYC6* and *APG9* levels were dramatically upregulated in the NAV and TDL isolates as compared to BCN 150 (Figure 6).

### 3.6. Both Isolates Exhibited a High In Vitro Infection Capacity

As part of the characterization of the new isolates, we aimed to compare the infection capability of such *L. infantum* isolates compared to the BCN 150 reference strain. The NAV and TDL isolates exhibited a higher efficiency for being phagocytosed by macrophages, infecting 72.4 and 72.5% of the host cells with 3.01 and 3.22 amastigotes per macrophage, respectively. Under the same conditions, BCN 150 infected 49.62% of the macrophages, with 1.97 amastigotes per infected cell (Figure 7).

### 3.7. Amphotericin B and Miltefosine Were Active against Intracellular Amastigotes

When infected cells were treated with 0.05 µM of amphotericin B, a reduction in the percentage of infected macrophages was observed in the three cultures (isolates and reference). Interestingly, after the treatment with 2 µM of miltefosine, significant decreases in the infection rate and the parasite burden were observed in the NAV and TDL isolates (Figure 8). At this concentration, no significant reduction was observed in cells infected with BCN 150.

## 4. Discussion

Climate change and increased travel and migration have contributed to the spread of leishmaniasis in Europe, which may allow the introduction of new exotic species of *Leishmania* [13] or change the profile of known strains [18]. This highlights the need to continue isolating and characterizing *Leishmania* strains from hosts. Here, two *Leishmania* isolates obtained from naturally infected dogs were characterized.

*L. infantum* is the most widespread *Leishmania* species in Spain. However, the presence of *L. donovani* in Europe has been reported, and its emergence in southern Europe is possible because of the diffusion of the vector [13] in various countries. Therefore, techniques for the discrimination between *L. donovani* and *L. infantum* are necessary.

Many genomic regions of *Leishmania* have been described as targets for molecular diagnosis, such as kinetoplast DNA (kDNA) [23,24], the internal transcribed spacer 1 (ITS1) [25], the mini-exon (ME) [23,26], heat-shock protein 70 (HSP70) [27] and 20 (HSP20) [28], and cysteine protease B (*cpb*) [15,21], among others. Nevertheless, only some of these allow the specific identification of the *Leishmania* species.

In this study, we used the methodology proposed by Hide and Bañuls (2006) [15], and Oshaghi et al. (2009) [21], and focused on a PCR restriction fragment length polymorphism (PCR-RFLP) for the specific species identification of the isolates. This methodology and the further sequencing of the amplified fragment, allowed us to confirm that we were working with *L. infantum* parasites, which were subsequently named as *L. infantum* NAV and *L. infantum* TDL. It was confirmed that this method is an effective method of *L. infantum* and *L. donovani* identification and discrimination. After the identification of both isolates, their characterization was performed and different genes were studied.

The gene expression levels of *CYCA*, *CYC6* and *APG9* were analyzed. Patino et al. recently observed that these three genes were upregulated in treatment resistant *L. amazonensis* [29]. In our case, *CYCA* was highly downregulated in both the NAV and TDL isolates. The inhibition of this gene has been previously associated with a negative regulation of cell proliferation [30], which was in agreement with the results obtained through the growth curve study. In fact, both isolates showed a lower growth rate than BCN 150. Nevertheless, the expression of *CYC6* and *APG9* was upregulated. *APG9* is involved in autophagy and cytoplasm-to-vacuole transport vesicle formation, and the upregulation of transcripts encoding the autophagy protein *APG9* has been associated with the recycling of protein under stress conditions and/or while undergoing a differentiation process [29].

Although the upregulation of the *CYC6* and *APG9* genes has been related to treatment-resistant parasites, the promastigotes forms of the isolates were significantly more susceptible to amphotericin B and miltefosine when compared to BCN 150. Moreover, the intracellular amastigotes of NAV and TDL were more sensitive to miltefosine treatment. The EC_50_ values for miltefosine evaluated in promastigotes of both isolates were 16.7 ± 3.3 µM (NAV) and 9.2 ± 1.7 µM (TDL), and were in a range (5.89–23.7 µM) previously reported by Espada et al. (2019) [31]. Our study indicated that isolates of *L. infantum* obtained from infected dogs in Europe did exhibit a similar susceptibility to miltefosine when compared to isolates from America.

Furthermore, recent studies have shown that the laboratory-adapted strains of *Leishmania* might carry some genomic mutations when compared to wild-type strains, due to successive in vitro passages [32]. It is well known that mutations can generate critical changes in key proteins. For example, the structure of such a protein and the efficacy of potential protein inhibitors might be different between the wild-type and laboratory strains. In our laboratory, after a high number of passages of isolate culture, mutations in *CYCA*, *CYC6* and *APG9* might therefore induce different responses to leishmanicidal drugs including miltefosine and amphotericin B.

In spite of their low growth rate, both isolates displayed a high in vitro infection capacity when murine macrophages were infected with stationary parasites at a low parasite/macrophage ratio. It would be interesting to further study if the aforementioned mutations might alter the growth rate as well as the infection capacity of the isolates.

## 5. Conclusions

*L. infantum* TDL and NAV are novel *Leishmania* isolates that might be useful for in vivo and in vitro assays that will allow a better understanding of the parasite biology and the disease in Mediterranean areas. In the future, more experiments need to be performed to analyze the genomic mutations detected after successive in vitro passages and to assess their impact on leishmaniasis outcomes.

## Figures and Tables

**Figure 1 animals-13-02153-f001:**
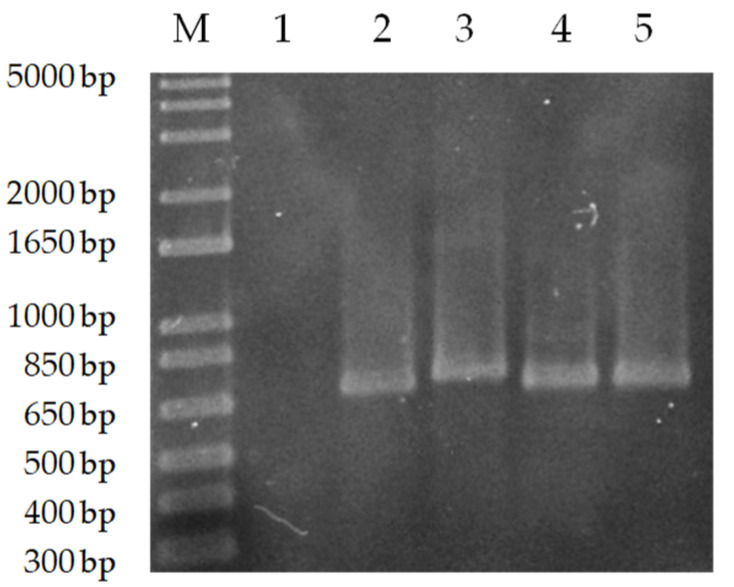
PCR amplification of the *cpb* gene from *L. donovani*, BCN 150, NAV and TDL genomic DNA. M: Marker (1 kb plus Ladder). 1: Negative control (*L. guyanensis* DNA), 2: Positive control (*L. infantum* BCN 150 DNA), 3: Positive control (*L. donovani* DNA), 4: NAV DNA, 5: TDL DNA.

**Figure 2 animals-13-02153-f002:**
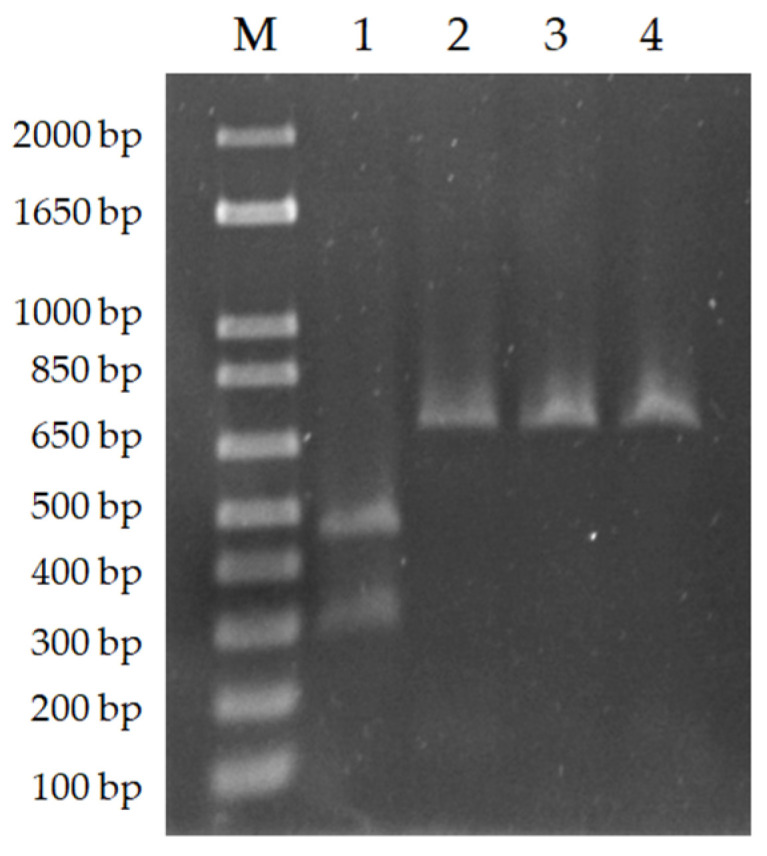
M: Marker (1 kb plus Ladder). 1: *L. donovani* PCR product digested with DraIII-HF (Positive control), 2: Non-digested BCN 150 PCR product, 3: Non-digested NAV PCR product, 4: Non-digested TDL PCR product.

**Figure 3 animals-13-02153-f003:**
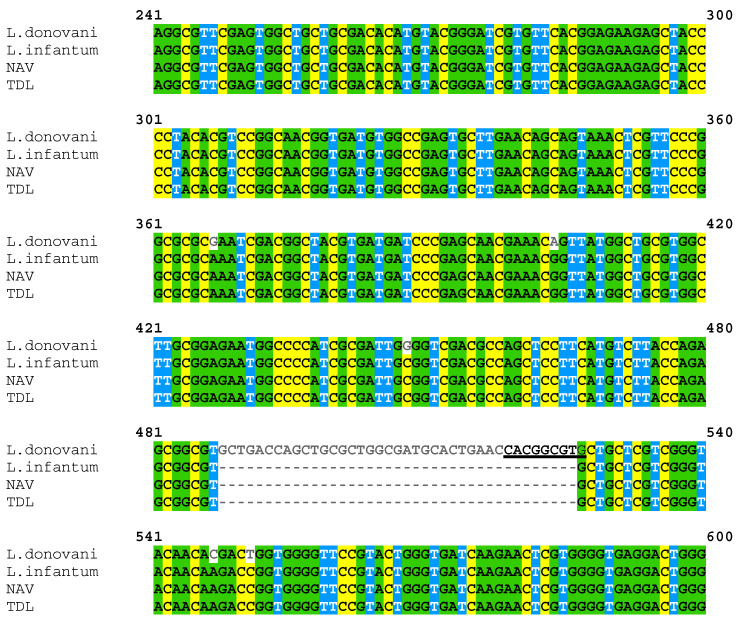
Alignment of the *cpb* gene sequences of *L. donovani* (AY896783.1), *L. infantum* (AY896777.1), NAV and TDL. The restriction target of the DraIII-HF enzyme was underlined.

**Figure 4 animals-13-02153-f004:**
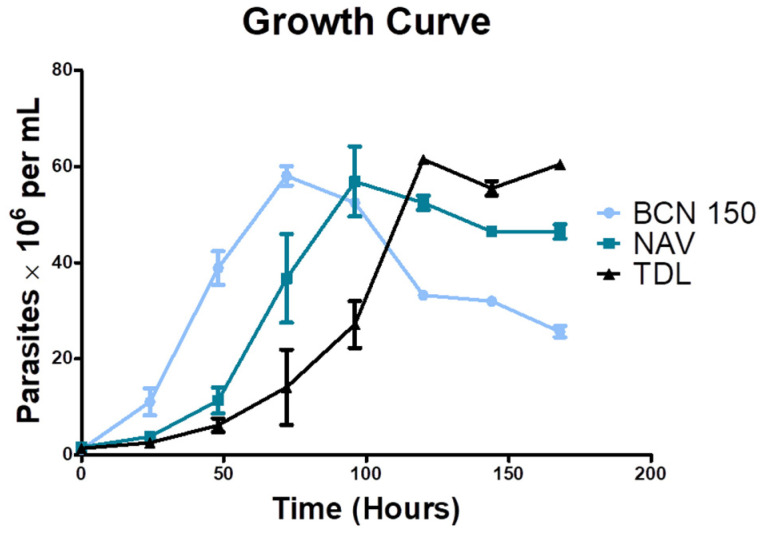
Growth curve of BCN 150, NAV and TDL promastigote parasites. Experiments were performed in sextuplicate, measuring parasite concentration every 24 h for 7 days. The results are expressed as the number of parasites per milliliter ± the standard deviations (SD).

**Figure 5 animals-13-02153-f005:**
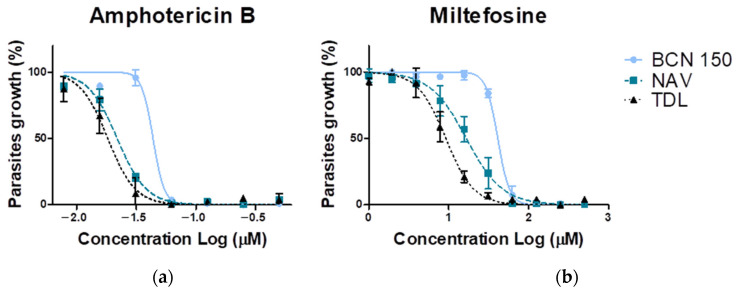
Dose–response curves of BCN 150, NAV and TDL promastigotes treated for 48 h at 26 °C with different concentrations of amphotericin B (**a**) and miltefosine (**b**). Experiments were performed in triplicate. The plots show means ± SD of parasite growth values measured at each concentration.

**Figure 6 animals-13-02153-f006:**
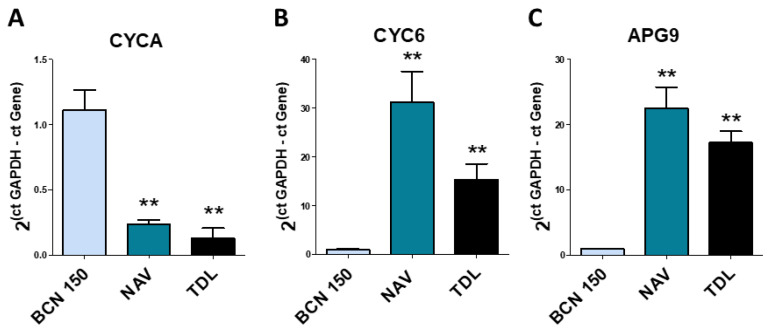
Gene expression analysis by quantitative real-time PCR in BCN 150, NAV and TDL. mRNA levels of the genes *CYCA* (**A**), *CYC6* (**B**) and *APG9* (**C**). Data are presented as gene expression ± SD normalized with *GAPDH* gene expression (** *p* < 0.01).

**Figure 7 animals-13-02153-f007:**
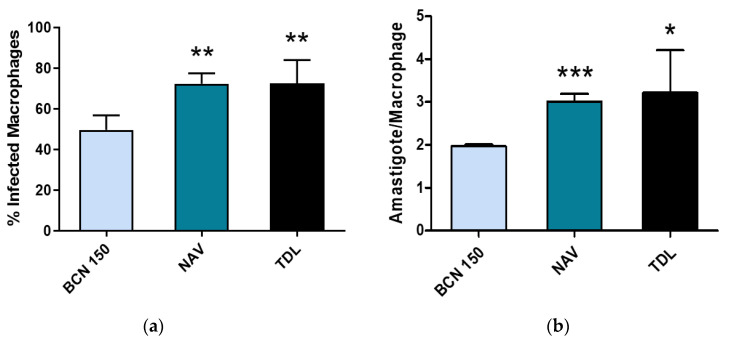
In vitro infection of murine macrophages with BCN 150, NAV and TDL. (**a**) Percentage of infected macrophages; (**b**) Number of amastigotes per infected macrophage. Data are presented as mean ± SD (* *p* < 0.1, ** *p* < 0.01, *** *p* < 0.0001).

**Figure 8 animals-13-02153-f008:**
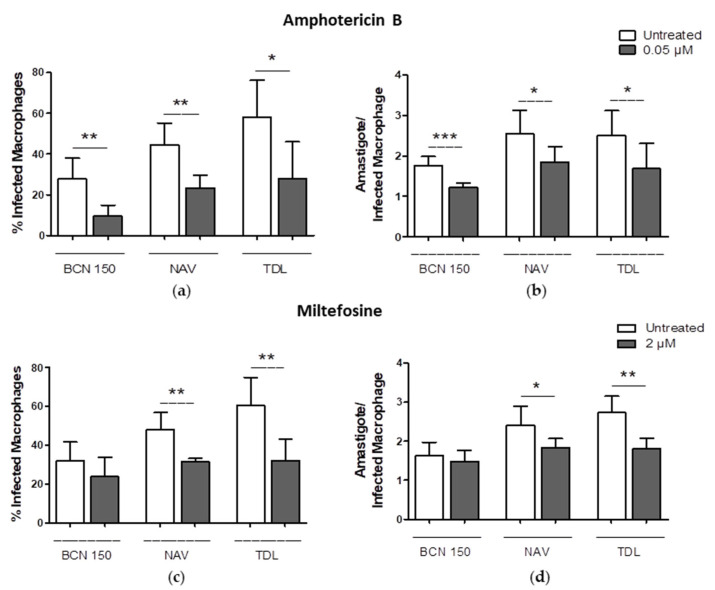
Effects of amphotericin B (**a**,**b**) and miltefosine (**c**,**d**) on isolate infection efficiency and parasite burden. The bars represent the means ± SD (* *p* < 0.1, ** *p* < 0.01, *** *p* < 0.0001). Results were obtained from two independent experiments.

**Table 1 animals-13-02153-t001:** Primers used for the study of gene expression.

Gene	Forward (5′→3′)	Reverse (5′→3′)
*GAPDH*	ACCACCATCCACTCCTACA	CGTGCTCGGGATGATGTTTA
*CYCA*	CCCCAACACCGCTGACTAAT	TCCGACTGGCGGCTCATGTA
*CYC6*	AGTACCCTGCACGCCTACTA	TTGTTGTTGGCGCAGGAAAG
*APG9*	TCACTCTCGTTTGGTGGCTC	AAAGGTCGTCGTGATGTGCT

**Table 2 animals-13-02153-t002:** EC_50_ of amphotericin B and miltefosine on BCN 150, NAV and TDL promastigotes 48 h after treatment (** *p* < 0.01).

Compound	BCN 150	NAV	TDL
	EC_50_ (µM)	EC_50_ (µM)	EC_50_ (µM)
Amphotericin B	0.043 ± 0.002	0.02 ± 0.01 **	0.018 ± 0.002 **
Miltefosine	41.1 ± 3.5	16.7 ± 3.3 **	9.2 ± 1.7 **

## Data Availability

The data presented in this study are available on request from the corresponding author for scientific purposes.

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
