# Peer review of "Characterization of Leishmania Parasites Isolated from Naturally Infected Mammals"

_animals, 2023, doi:10.3390/ani13132153_

Round 1
Reviewer 1 Report
Title: Characterization of Leishmania parasites isolated from biological samples of naturally infected mammals
Overview: Burguete-Mikeo and colleagues present an article whose aim of the work was to isolate and characterize two Leishmania isolates (NAV and TDL) obtained from naturally infected mammals.
The idea of the manuscript is to identify new circulating Leishmania strains to understand their differences from standard strains, which would help many research works. However, some concepts need to be revised and changes should be made to the text.
1) It needs to be placed in the text in which mammal the strains were isolated (Human? animal?). This information is very important and is nowhere to be found. Not the methodology. It needs to appear from the summary to the end of the text.
2) I recommend putting in the abstract what these genes are related to. (Lines 35-37)
3) Line 84 , line 362- Make it clear where the strains were isolated from.
4) Lines 127-128 – has two concentrations of hepes in the culture medium. put which one was used.
5) Line 145 - what drug concentrations? How was the choice of these concentrations based?
6) Line 178 - What type of mouse was used? balb/c? c57bl6? Put in the text which mouse specie was used.
7) Line 177 - what protocol to differentiate bone marrow macrophages was used? GM-CSF? fibroblast cell line supernatant?
8) Line 312 - The letter A in figure 6 is CYCA. Please correct.
9) Line 355 - put in the figure and in the caption that the gray bar refers to the group stimulated with the drug. This information is not included.
10) what is the drug used where these strains were isolated? it would be good to comment on this in the introduction even to justify the use of these drugs in the experiments.
11) Line 373 - … proposed by …
12) Line 391 - As these two genes are related to resistance to treatment wouldn't you expect the result to be downregulated? since they are susceptible to the drugs used? How would you explain these results? Are these genes related to any specific drug?
Minor editing of English language required.
Reviewer 2 Report
The study is presented very well, the methods are laborious and the results are interesting. However, consider that some improvements should be made in terms of the discussion of results.
Section 2.5 is not well understood
Why use the MTT method instead of the more common acid phosphatases method?Where was the amphotericin and miltefosine tested from obtained? In what did they dissolve these drugs?
Did they not determine the selectivity index of the drugs in the parasite?
Please, I expanded the conclusion, there is much more to place
In the discussion, from lines 392 to 394 it is necessary to compare the values obtained with those found by other authors.
Add a photograph of the wounded amastigotes and of the amastigotes treated with amphotericin B and miltefosine
For what purpose did you test miltefosine and amphotericin B in promastigotes and amastigotes? I do not see it implicit in the fundamental objective of the work.
I believe that the discussion can be expanded, they have interesting results that can be compared with other studies.
Is necessary review gramatical aspects of language
Reviewer 3 Report
Considering the type of work, this manuscript would be more appropriate to be submitted as a Communication.
Title – adapt to read as: Characterization of Leishmania parasites isolated from naturally infected mammals
Simple summary – please mention the host species from which the parasites were isolated
Abstract – the same comment as above
Keywords – display alphabetically
Line 47 – 20 (instead of twenty)
Line 49 – aspects
Line 51 – 98 regions or 98 countries?
Line 53 – delete “rate”
Line 54 – please update reference (and mortality)
Line 56 – Leishmania spp. [spp. in non-italic type]
Line 63 – replace asymptomatic with apparently healthy
Line 74 – write Leishmania infantum at its first use, then L. infantum – it is not necessary to use Leishmania infantum (L. infantum) – the same comment for the other species
Line 75 – (L.) [subgenus Leishmania] does not seem to be necessary; if the authors want the present this subgenus, they should write out as (Leishmania) at its first use
Line 76, etc. Leishmania spp. (instead of Leishmania species) – change accordingly throughout the manuscript
Line 83 – which infected animals were these? And what type of samples were those?
Line 104 – L. guyanensis
Line 230 – adapt as: … belong to species L. infantum within L. donovani complex
Discussion/Conclusions – please elaborate on the possibility that strain characteristics change with culture time, etc.
English would benefit from a review, preferably by a native speaker of the language.
For example:
The climate change and the increase of travels and migrations, contribute to the spreading of leishmaniasis in Europe, and may allow the introduction of new exotic Leishmania species or the alteration on the profile of known strains.
Would read better as:
Climate change and increased travel and migration contribute to the spread of leishmaniasis in Europe, which may allow the introduction of new exotic species of Leishmania or change the profile of known strains.
This reviewer will not point all the less accomplished aspects of English.
